# Computational Structural Biology: Successes, Future Directions, and Challenges

**DOI:** 10.3390/molecules24030637

**Published:** 2019-02-12

**Authors:** Ruth Nussinov, Chung-Jung Tsai, Amarda Shehu, Hyunbum Jang

**Affiliations:** 1Computational Structural Biology Section, Basic Science Program, Frederick National Laboratory for Cancer Research, Frederick, MD 21702, USA; tsaic@mail.nih.gov (C.-J.T.); jangh2@mail.nih.gov (H.J.); 2Sackler Institute of Molecular Medicine, Department of Human Genetics and Molecular Medicine, Sackler School of Medicine, Tel Aviv University, Tel Aviv 69978, Israel; 3Departments of Computer Science, Department of Bioengineering, and School of Systems Biology, George Mason University, Fairfax, VA 22030, USA; amarda@cs.gmu.edu

**Keywords:** big data, machine intelligence, bioinformatics, biological modeling, free-energy landscape, mutations

## Abstract

Computational biology has made powerful advances. Among these, trends in human health have been uncovered through heterogeneous ‘big data’ integration, and disease-associated genes were identified and classified. Along a different front, the dynamic organization of chromatin is being elucidated to gain insight into the fundamental question of genome regulation. Powerful conformational sampling methods have also been developed to yield a detailed molecular view of cellular processes. when combining these methods with the advancements in the modeling of supramolecular assemblies, including those at the membrane, we are finally able to get a glimpse into how cells’ actions are regulated. Perhaps most intriguingly, a major thrust is on to decipher the mystery of how the brain is coded. Here, we aim to provide a broad, yet concise, sketch of modern aspects of computational biology, with a special focus on computational structural biology. We attempt to forecast the areas that computational structural biology will embrace in the future and the challenges that it may face. We skirt details, highlight successes, note failures, and map directions.

## 1. Introduction

Computational biology has made vast strides. Rather than being a ‘second fiddle’ to experiments, it now often leads research. When the number of candidate experimental targets is daunting, computational biology can come to the rescue to filter, prioritize, and provide data-based hypotheses and leads. Vast and heterogeneous data are increasingly accessible, and computational biology can efficiently integrate these diverse information-rich resources, evaluate, and interpret the outcomes [1,2]. With massive genomic, transcriptomic, and proteomic data, as well as structural foot printing, computational biology has also made great strides toward a more reliable multiscale biological modeling [3]. In addition, it has developed software for inferring molecular interactions and assembling them into interconnected cellular pathways [4,5,6,7,8,9,10,11]. Steps have also been taken toward the modeling of cells, gearing to blend molecular, cryo-electron microscopy (cryo-EM), cryo-electron tomography (cryo-ET), cellular, and systems/human scales [12,13,14,15,16] and to facilitate in situ structural biology studies on a proteomic scale [17]. Inspired by Da Vinci’s imagination, a symposium has even been organized on *modeling and imaging the whole human body at the atomic scale to understand the human body* (https://dornsife.usc.edu/bridge-at-usc-bak/da-vinci-symposium/). Computational biology has successfully identified disease-linked genes [18,19,20] and harnessed artificial intelligence neuron connectivity and electrical flow to model the brain. The sequencing of individuals has permitted comparisons of corresponding sequences in diseased and healthy tissues, and with the help of computational biology, technological advances have accomplished the imaging and tracking of molecules in action in single cells [21,22,23]. Network science has prospered and become widely used [24] in applications ranging from signaling networks in the cell to those regarding protein molecules in allosteric communications [25,26,27,28,29,30,31,32,33,34,35,36,37,38,39,40,41,42,43,44]. Compelling advances have also been made in modeling protein and RNA structures and in mapping chromatin and its dynamics at high resolution [45,46,47,48,49,50,51,52]. These advances are compelling since, despite the high-throughput data, understanding cell signaling networks is listed among the top unanswered questions of modern science. Computational biology has also taken up the complexity of diseases to understand their mechanisms, systemic behaviors, and linkages within an organism as well as epidemiology across populations. Computational and mathematical modeling of complex biological systems has flourished [53,54], and impressive progress has been made in synthesis and nanobiology. As a result, now computational biology is spearheading microbiome research. All this has been possible thanks to the vast advances in computing power (albeit still not enough) and machine architectures. Recently, we have commented on the advancements and challenges in computational biology [2,55]. As the references above indicate, the last 4–5 years have already seen shifts and giant leaps forward, especially with respect to the harnessing of big data and machine intelligence [56].

In line with the aim of this Special Issue, here, we focus on computational structural biology. It is convenient for scientists to consider biological molecules in terms of their sequences. Such a simplification bypasses the challenge of reliably modeling their structures on a large scale under diverse conditions and accounting for their function-related fluctuations. However, in reality, *no molecule exists in the cell as a mere string with covalently linked chemical blocks*. Biological macromolecules fold as they are being synthesized either into stable three-dimensional shapes or into multiple interconverting states to populate an ensemble of ‘intrinsically disordered’ states [57,58,59]. Thus, computational structural biology that considers these states—which are what the cell ‘sees’—is of fundamental importance, even though, sometimes, shoved to the sidelines to permit simplification and faster analysis. Here, our discussion initiates with computational biology and then moves on to computational structural biology, formulating what we perceive could be its directions in the future.

## 2. The Breadth of Computational Biology

Bob Murphy, the head of the Computational Biology Department at Carnegie Melon University (http://www.cbd.cmu.edu/about-us/what-is-computational-biology/), frames computational biology by asking the broad question of how to “learn and use models of biological systems constructed from experimental measurements”. The aim is not necessarily to increase the understanding of the system, which may be so complex that it may not be fully understood or predicted; instead, the aim can be the creation of the model itself, even if currently unproven. Nevertheless, as scientists, our quest is always to understand.

Areas encompassed by computational biology depend on the specific goals and the types of experimental data that are available, including sequence and structural analysis [60,61] and their correlation with function [62], evolution and population genomics, regulatory [63] and metabolic networks [64], image analysis [65], and disease [66]. Computational biology often addresses tasks by analyzing large genomic [67,68,69,70,71], proteomic, microarray, cell and tissue imaging, and clinical data [72,73] to produce robust statistical trends and correlate these with outcomes. It harnesses high-throughput genomic and proteomic methods to integrate data [74], to identify and validate biomarkers and novel therapeutic targets, and to rapidly translate the findings to the clinic. Recently, it has also made great strides in assembling many different data types, revolutionizing the impact and power of biological information [75]. Statistics reflect trends and correlations in the *population*. However, because the number of entries is limited, statistics are not able to provide a detailed description including co-occurrences of multiple features for an individual entity; thus, statistics are unable to explain trends and observed biases [76], which may restrict the predictive power for individual patients [77]. To understand observations, there is a need to look at individual molecules or cells. At the other end of the spectrum are experimental approaches, which are often less quantitative and less detailed. Typically, studies deal with molecules, pathways, cells, and tissues. Crystal structures, and recently their cryo-EM images, are often presented; however, their ensembles are often overlooked, as are considerations of how different environmental conditions affect conformational distributions, which in turn determine the molecular functions.

## 3. The Quest to Understand the Molecular Mechanisms

A structural biology approach is based on the free-energy landscape. “Population shift”, or the redistribution of the conformational substates in the ensemble, links basic physicochemical principles to physiological functions and dysfunctions in disease [78]. This conformational view forms the basis for many computational structural biology projects, leading to their distinctiveness and to conceptual innovative advances. Computational structural biology can be harnessed to reveal the mechanisms of mutations and signaling specificity [79]. The outcome illustrates that computational structural biology may reach a level of mechanistic detail that is hard or impossible for experiments or nonstructural approaches to attain. Cancer treatment decisions would benefit from marrying statistical analysis with genetic and the structural analyses. Together with experimental approaches, statistical ‘big data’, genetic, and clinical analyses, and fundamental theory, computational structural biology can help coin and establish new paradigms to elucidate the basis of cancer and innovate treatments.

Computational biologists are driven by a fundamental ‘quest to understand.’ They exploit ‘big data’ and statistical trends for leads and search the experimental literature for functional observations. When possible, they cross-feed with experimental collaborators. This approach spawns more robust prediction schemes and allows reconciling experimental observations. It additionally leads to deeper understanding and innovative ideas, such as, for instance, the role of calmodulin in phosphatidylinositol-4,5-bisphosphate 3-kinase α (PI3Kα) activation in oncogenic KRas signaling and signaling selectivity at the membrane [35,80,81,82,83]. The structural view also underlies innovative interface-based pathogen–host protein interaction prediction methods [84]. Distinct from others, the approach in reference [58] is not based on the protein sequence nor on its entire structure, but on binding interfaces, which adopt favored motif architectures. This focus also guides studies of pathogen target-specific recognition, as well as other projects, including allostery in the T cell receptor (TCR)–CD3 complex (with crystallographers/NMR collaborators [85]), amyloid seeding [86], and recognition by homologous antibodies [87]. Other examples include elucidating the conformational heterogeneity of the ATP-binding cassette (ABC) transporter and its functional relevance [88], the crystal structure of the C2 domain of PI3Kα in complex with the phosphoinositide head-group mimic inositol hexaphosphate, revealing two distinct pockets for membrane binding [89] and a replacement of moieties inducing a molecular switch which transforms the molecule from a negative allosteric modulator of a receptor into an activator of Wnt signaling [90]. Additional examples with strong wet-lab components include disulfide tethering of a non-natural cysteine (KRas^M72C^) identifying a new Switch-II pocket binding ligand (2C07) of the active GTP-bound state, transforming the pocket to that observed in the GDP-bound state [91], and experimentally confirming that KRas populates conformational states different from its isoform HRas and the oncogenic mutant KRas^G12D^ [92], as can be expected considering that the sequences are not identical. Further examples that combine experiments with computations include uncovering the isoform-specific signatures in Ras interactions [93], redefining the protein kinase conformational space with machine learning [94], developing covalent inhibitors of epidermal growth factor receptor (EGFR) [95], proposing c-Jun N-terminal kinase (JNK) signaling as a therapeutic target for Alzheimer’s disease [96,97], describing bacterial Ras/Rap1 site-specific endopeptidase cleavage of Ras disrupting Ras/extracellular signal-regulated kinase (ERK) signaling through an atypical mechanism [98], and finally, the remarkable observation of the biased antagonism of the CXC chemokine receptor type 4 (CXCR4) [99]. Additional examples, not detailed here are described in other works [34,100,101,102,103,104,105,106,107,108,109,110,111].

## 4. Challenges in Computational Structural Biology

Intertwined with computational biology, computational structural biology has continued to chart its path. After 50–60 years, the problem of predicting the three-dimensional structures of proteins from their sequences remains unsolved. With the apparent stalling in the progress of the ‘true’ ab-initio folding from ‘first principles’, focus has shifted to three components in modeling algorithms: the energy function, the conformational search, and the model selection [112,113,114,115,116]. Great progress was reported recently by AlphaFold, a deep learning approach by Google’s DeepMind team that outperformed other teams for about half of the targets in the 2018 Critical Assessment on protein Structure Prediction (CASP) community-wide competition [117]. Yet, even the impressive performance of a two-year effort by Google’s dedicated team of AI and machine learning researchers failed on more than half the targets and considered, like most approaches, a narrow version of the structure prediction problem. To appreciate the narrow context, it should be noted that, broadly, computational structural biology deals with structures of biological macromolecules and their interactions not only with each other but also with water, ions, lipids, or small molecule effectors in solution or at (on or in) the membrane, and with the consequences of their modifications and mutations. Especially, computational structural biology aims to model and exploit the structural landscape to understand protein function and dysfunction by harnessing the active and inactive states and considering the shape of the free-energy landscape, identifying the conformations at its minima, the metastable states, and the barriers which need to be crossed to switch between the states [118,119,120,121,122]. It also considers how mutations and other events alter the landscape, by populating previously hidden states. Structural dynamics, which embraces fluctuations between states, is essential for functional elucidation. Any structure-based study needs to consider the complexity of the environment and its impact on the structural dynamics of any uncomplexed or complexed molecular system. This is no short order, even for the most sophisticated machine learning approaches.

## 5. Some Emerging Principles in Computational Structural Biology

Among the tenets that computational structural biology may increasingly embrace in the next few years are integrated deep neural networks, Markov state models, and metastable states for sampling conformational space, which will permit mining the entire free-energy landscape, thus providing insight into biological macromolecular actions, including catalysis [123,124,125,126,127,128,129] and dynamic networks [130]. Other topics of interest are integrative or hybrid modeling across disparate scales [131,132,133,134,135,136], including organelles, cells, and tissues, and archiving of the models [137]; harnessing machine intelligence to extract trends and predict outcomes; making headway in precision medicine [138,139]; improving software for imaging technologies and analysis and unveiling the mechanisms, on the structural level, through which the microbiota can hijack host signaling and impact human health. Finally, integration with experimental structural data, such as cryo-EM and spectroscopy at different scales, are emerging as key to the successful modeling of multimolecular assemblies [3].

There are also older topics which are still awaiting a solution. A central problem in molecular dynamics simulations is that it is not possible yet to compute the free energy from standard simulations (i.e. from end-point simulations). Even enhanced sampling methods providing free-energy landscape are not able to dissect the contributions to free energy. Similar to other free-energy methods, these are very slow. Entropy can be now accurately addressed for solutes by the k^th^ nearest-neighbor and maximum information spanning tree method, but still, solvation entropy is not computable from standard simulations. Implicit solvent methods could play an important role in this field. Indeed, with more refined solvation models, several proteins could be correctly folded (as shown by the Simmerling group [140,141]). Free-energy calculations are a vastly important and challenging open problem. Much effort has also been invested in forcefield development, but there is still much room for further improvement.

## 6. Areas that May Take the Center Stage

Below, we list some of the areas that we foresee as taking the center stage and gaining momentum in the next years. In many, machine intelligence and an increasing level of automation are expected to become methodological requisites. In particular, we posit that a combination of deep neural networks (DNN) and Markov state models (MSM) is applicable to most of the topics on the list:(a)Modeling large molecular assemblies and critically figuring out their assembly–disassembly processes in the cell to regulate its functions(b)Modeling chromatin structure and dynamics and, especially, figuring out their regulation(c)Regulation of signaling in key protein nodes and between them in the cell(d)Modeling and prediction of drug resistance(e)Integration of experimental statistical ‘big data’ and the structural landscapes to model cells (tissues) behavior and system complexity(f)Precision medicine, to identify and predict drug targets, and drug discovery(g)Figuring out how the microbiota hijacks cell signaling and cell response to infection(h)Efficient sampling of the conformational space(i)Modeling across scales(j)Figuring out molecular mechanisms in detail and how these are commandeered by mutations in disease(k)Untangling redundant signaling pathways in the cell(l)Designing functional molecules and cells(m)Generating detailed, high-fidelity, synthetic biological data in silico to test hypotheses and advance model building, testing, and biological knowledge.

## 7. Conclusions

Bioinformatics is undergoing a revolution. Traditional statistical approaches increasingly give way to advanced algorithms. Advances in machine learning have been shown capable of representing potential-energy surfaces by fitting large data sets from electronic structure calculations [142], and a ‘Machine Learning in Health and Biomedicine’ collection was conjointly published in *PLOS Medicine*, *PLOS Computational Biology*, *PLOS ONE* (https://collections.plos.org/mlforhealth), and other journals [143], illustrating the usefulness and diversity in bioinformatics’ applications toward improving human health. This is coupled to the vast increase in the generation of data and computational power, without which machine learning cannot be reliably executed. Machine learning-based methods are powerful, and their comparisons with the more traditional strategies illustrate their advantages. Are these going to replace the traditional approaches? Biology has long strived to shift from a descriptive to a quantitative science. However, the increasing availability of data—due to automation in experimental approaches—is leading to a paradigm shift in computational biology, forcefully pushing biology not only from a descriptive to a quantitative science but also from a descriptive to an automated science.

Nonetheless, the hallmarks have not changed. The key is to solve the questions that are still unanswered. The quest is to understand observations at the detailed level and to predict them. The paradigm underlying computational structural biology argues that to truly understand, one must have knowledge of the structure. Computational structural biology is a vast field. In this review, large areas of research are only sketched, and some are altogether missing. Our aim is to indicate highly important tasks that can be addressed by structural modeling and simulation and can thus be inspiring for the readers. Examples are provided to show that the methods and computational power are (and will be more and more) adequate for the tasks listed.

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
