# Peer review of "Computational Structural Biology: Successes, Future Directions, and Challenges"

_molecules, 2019, doi:10.3390/molecules24030637_

Round 1

Reviewer 1 Report

Computational biology has made a giant leap during the last couple of years. It has been transformed from a tool that helps to understand, sort out and explain experimental data to the one that precedes the experiments and in reality helps to design them. Today the amount of data generated every day is sky-rocketing and processing them requires both more efficient computers and totally different algorithms.

In the manuscript the authors have tried to describe the task of modern computational biology in a very condensed way. I do not know their original aim, I mean whether the paper should be rather general or more descriptive. If I wrote such a paper I would prefer it to be more detailed with a couple of concrete examples. On the other hand, the manuscript contains numerous references where the reader can find a lot of examples. If the aim was to write a very general paper about modern aspects of computational biology, the manuscript really fulfils the task and, according to my opinion, it can be published as it is.

I have managed to find just one typographical error:

l. 58:    synthetic should be replaced by synthesis   

Author Response

We thank the reviewer for pointing out the typo, which we have now fixed. In particular, we clarified the focus of the Perspective, including the Title, Abstract and throughout the text and streamlined it.

Reviewer 2 Report

Overall this is a concise review trying to review an extremely wide and diverse field. Computational biology now ranges from image recognition, to brain modeling, genome analysis up to modern structural biology.  Perhaps, a clearer focus in the introduction on the actual topic, computational, structural biology would have been better.

The introduction  contains uite a few grand statements in the introduction, that may need to be backed up by references, in line 44 for example  the claims that diseases-linked genes have been identified, harnesses artificial intelligence and electric flow to model the brain. The first statement clearly needs a concrete example and a reference, while I fail to understand what the authors are trying to say.

Likewise, in line 57 it remains unclear what the ‘impressive progress’ in synthetic and nano biology are, and how computational biology has contributed more than a simple tool.  The reference to ‘machine intelligence is rather meaningless at the end of the paragraph

When it comes to computational structural biology,Considering how central crystal structures and more recently EM structures are in the molecular understanding of biology I am not are of one can argue that these are ‘sometimes' presented. Clearly, macromolecular crystallography and cryo electron microscopy had their fair share of Nobel prizes which could have been better reflected. In addition, the work by Karplus and other to develope the modern force fields that are today the cornerstone of some many computational structural projects could have been mentioned.

Nevertheless, overall, this is a nice review and should be published, possibly with some addtion as outlined above

Author Response

The introduction contains quite a few grand statements in the introduction, that may need to be backed up by references, in line 44 for example the claims that diseases-linked genes have been identified, harnesses artificial intelligence and electric flow to model the brain. The first statement clearly needs a concrete example and a reference, while I fail to understand what the authors are trying to say.

We have inserted numerous new references in the Introduction, and as noted in our response to reviewer #1 have edited it extensively, aiming to further clarify and focus it.

Likewise, in line 57 it remains unclear what the ‘impressive progress’ in synthetic and nano biology are, and how computational biology has contributed more than a simple tool.  The reference to ‘machine intelligence is rather meaningless at the end of the paragraph

We used this terminology aiming to convey our belief that computational biology (like all computational sciences) has benefitted immensely from such advanced algorithms, and the field will increasingly employ it. We hope that within the framework of the revised version it is clearer.

When it comes to computational structural biology, Considering how central crystal structures and more recently EM structures are in the molecular understanding of biology I am not are of one can argue that these are ‘sometimes' presented. Clearly, macromolecular crystallography and cryo electron microscopy had their fair share of Nobel prizes which could have been better reflected. In addition, the work by Karplus and other to develope the modern force fields that are today the cornerstone of some many computational structural projects could have been mentioned.

In the original version we did not detail experimental methodologies, as this was not our aim. However, following the comments by the reviewer we have now explicitly related to it. As to forcefields, again, we related to improvements in simulations across scales which certainly require reliable forcefields, but now mentioned this specifically.

Reviewer 3 Report

The review by Nussinov and co-workers addresses the ambitious tasks that 

computational structural biology should face in the (next) future.

The main merit of this review is to indicate high level tasks which can be 

addressed by structural modeling and simulation, and thus can be inspiring for 

readers. Examples are provided to show that the methods and computational power 

are (and will be more and more) adequate for the tasks listed. 

The main limitation of the review is that it is very short compared to the 

subject, and therefore large areas of research are only sketched. As a 

consequence the many methodologies and applications are mentioned in the paper 

in a very dense way, whereas they would likely deserve more space.  

I have only a specific comment. Besides the topics listed by the authors as 

areas that will be addressed in the next future, I believe there are also more 

traditional, perhaps less exciting, topics which are still awaiting a solution. 

The authors mention successes, but also the existing difficulties in structural 

predictions. Similarly, successes/difficulties are found in other areas. 

I believe a central problem in MD is that still it is not possible to compute 

the free energy from standard simulations (i.e. from end-point simulations). 

Even enhanced sampling methods providing free energy landscape are not able to 

dissect the contributions to the free energy. Similar to other free energy methods these 

are very slow methods. Entropy can be now accurately addressed for solutes by the kth 

nearest neighbour and maximum information spanning tree method, but still 

solvation entropy is not computable from standard simulations. Implicit solvent 

methods could play an important role in this field. Indeed with more refined 

solvation models several proteins could be correctly folded (e.g. by the group 

of Simmerling). I believe the authors could consider at least to mention free energy calculation 

(which is truly a large area of research) as an open problem. 

Author Response

We thank the reviewer for the excellent comments. Along the line suggested by the review, we have added a new section to the revised manuscript.